# Spatiotemporal development of late and moderate preterm infant gut and oral microbiomes and impact of gestational age on early colonization

Sinéad Ahearn-Ford,[1] Andreas Kakaroukas,[2] Gregory R. Young,[1] Andrew Nelson,[3] Marieke Abrahamse-Berkeveld,[4] Ruurd M. van Elburg,[5] Darren Smith,[3,6] Janet E. Berrington,[1,2] Nicholas D. Embleton,[2,7] Christopher J. Stewart[1]

**ABSTRACT**   Microbiome research focusing on late and moderate preterm infants (LMPT; 32 to 36 weeks gestation) is limited, despite rising LMPT births, large healthcare burdens, and increased risks of multiple morbidities, potentially microbially related. In this longitudinal cohort study, 16S rRNA gene sequencing was used to analyze 371 stool and 402 saliva samples from 160 LMPT infants, collected at five time points between birth and 12 months corrected age (CA), to describe spatial and temporal variability in gut and oral microbiomes. Paired stool and saliva samples ($n = 337$) were analyzed for potential microbial relationships. Early LMPT samples (up to 60 days of life; DOL) were also compared with data from seven extremely preterm infants (EP; <28 weeks gestation; stool $n = 14$, saliva $n = 14$). LMPT stool and saliva were composed of distinct microbial communities at each time point, and both sample types showed increasing alpha diversity over time. Stool was initially dominated by *Escherichia/Shigella*, *Klebsiella*, and *Streptococcus*, with *Bifidobacterium* becoming dominant from term equivalent age (TEA). Contrarily, saliva was dominated by *Streptococcus* throughout the first year, with early contributions from *Staphylococcus* and later *Veillonella*. LMPT infants had higher stool and lower saliva diversity compared with EP infants. Both sample types from EP infants were taxonomically distinct from LMPTs, with *Escherichia/Shigella* dominating both EP sample types throughout the first 60 DOL. The results highlight the unique trajectories of LMPT microbiomes and emphasize the role of gestational maturity in shaping microbial communities.

**IMPORTANCE**   The oral and gut microbiome develops from birth and plays important roles in health. This has been well studied in extremely preterm infants (EP; born <32 weeks gestation) and term infants (born >38 weeks gestation), but there is a paucity of research describing oral and gut microbiome development in late and moderate preterm infants (LMPT; 32 to 36 weeks gestation). Our study analyzed microbiome development in 160 LMPT infants from birth to 12 months corrected age. The results showed distinct microbial communities in stool and saliva, with increasing alpha diversity and niche specification over time. LMPT infants' gut microbiome became dominated by *Bifidobacterium* by month 3, while the oral community was consistently dominated by *Streptococcus*. These results highlight that LMPT infants have gut and oral microbiome development that is more like term infants than EP infants, which has important implications for the care of LMPT infants.

**KEYWORDS**   preterm infant, gut microbiome, oral microbiome, spatiotemporal microbiome development

Address correspondence to Christopher J. Stewart, christopher.stewart@newcastle.ac.uk.

N.D.E. and J.B. declare research grants from Danone Early Life Nutrition, NeoKare UK, and Prolacta Bioscience paid to their institution in the last 5 years. N.D.E. declares lecture honoraria from Nestlé Nutrition Institute and Abbott Nutrition and declares providing consultancy advice to legal firms involved in class action for infants developing NEC; all honoraria and consultancy fees were donated to charity. M.A.-B. is an employee of Danone Nutricia Research, which funded the study. R.M.V.E. was an employee of Danone Nutricia Research until 2020. All other authors reported no conflicts of interest.

See the funding table on p. 13.

Late and moderate preterm infants (LMPT; born between 32 to 36 weeks of gestation) represent the majority of infants born preterm and account for 6%–7% of all births in England and Wales (1–3). Preterm infants are subject to multiple exposures, including prolonged hospitalization, antibiotics, and medical interventions that can interfere with microbiome development, characterized by delayed colonization of commensals and increased abundances of pathobionts (4, 5). Compared with those born full-term, LMPT infants are predisposed to a range of adverse short- and long-term health outcomes, including respiratory, metabolic, and gastrointestinal disorders, neurodevelopmental and socioemotional impairment, among others (6, 7).

The importance of the gut microbiome during infancy is well established, playing roles in essential processes, such as digestion, immune function, and metabolism (8). The oral cavity contains the second most diverse microbial community of the human body after the gut and harbors a distinct community of microbes, which may colonize the gut and have been associated with various oral and systemic diseases (9). To date, most studies have focused on gut microbiome development in very and extremely preterm infants with severe health challenges (10–15). Few studies have addressed late and/or moderate preterm infants in descriptions of gut or oral microbiome development, and most that have, have been limited by small sample sizes, sparse sampling, or short windows of observation (16–23). Of these, the most extensive has involved the characterization of 41 LMPT infants' gut microbiome development up to 1 year (20), and the development of the oral microbiome in 21 human-milk fed moderate preterm infants up to 210 days after birth (16).

The current study aimed to comprehensively describe the simultaneous development of gut and oral microbiomes in a cohort of 160 LMPT infants during the first year of life. Additionally, early development of both ecosystems was compared with those from an existing cohort of 7 extremely preterm infants (<28 weeks gestation) (15) to further explore the influence of gestational maturity.

## MATERIALS AND METHODS

### LMPT patient cohort and study design

LMPT (born between 32 and 36 weeks of gestation) infants included in this analysis were cared for in the Royal Victoria Infirmary, Newcastle upon Tyne, United Kingdom (UK) and recruited to the Feeding Late and Moderate Preterm Infants Nutrition and Growth Outcomes (FLAMINGO) study between May 2018 and March 2020 (registration number: ISRCTN15469594) (1). Infants with medical problems likely to affect growth or nutrition, including genetic disorders, congenital anomalies, and heart defects, were excluded. The cohort included 20 pairs of twins and two sets of triplets.

Samples were collected by nurses, parents, or guardians longitudinally at five study visits: Entry (the time of enrollment into the study), Term Equivalent Age (TEA; the gestational age at which an infant is considered full-term, i.e., 40 weeks in this study), and at 3, 6, and 12 months corrected gestational age (CA; the age the infant would be if born at full-term, calculated as chronological age minus the number of days born preterm). Stool and saliva samples acted as proxies for gut and oral microbiomes, respectively. Stool samples were obtained from nappies in sterile universal containers. Saliva samples were collected using sterile, polyester, DNA-free, DNase-, and RNase-free swabs (PurFlock Ultra 6' sterile DNA-Free) of oral mucosa and sealed in dry polypropylene tubes. Biological samples were briefly (maximum 7 days) stored at −20°C before being transferred to −80°C.

### 16S rRNA gene sequencing and taxonomic profiling

DNA was extracted from all available stool ($n$ = 390) and saliva ($n$ = 413) samples using the DNeasy PowerLyzer PowerSoil Kit (QIAGEN). DNA was extracted from ~0.1 g of stool, and DNA was extracted from saliva using the entire collection tips of oral swabs

(detached using the breakable feature at the junction between the shaft and the tip). Two negative kit controls (containing no biological sample) were extracted for every batch of 96 samples ($n = 18$). Owing to the relatively low microbial biomass of saliva samples, the following modifications were made to the extraction protocol for these samples: reducing the volume of PowerBead solution added to 500 µL and combining solution C2 and solution C3 (1:1).

Sequencing and bioinformatic processing of the variable region 4 (V4) of the 16S rRNA gene were performed by NU-OMICS (Northumbria University, Newcastle upon Tyne, UK) based on the Schloss wet-lab MiSeq SOP (24). Briefly, PCR was carried out using 1× KAPA2G Robust HotStart ReadyMix, 0.5 µM each primer (515f, 806r [25]) under the following conditions: 95°C 2 min, 30 cycles 95°C 20 s, 55°C 15 s, and 72°C 5 min with a final extension 72°C 10 min. Fastq files were processed using Mothur (v1.48) (26), merged, and filtered to remove low-quality reads. Sequences were aligned to the Silva reference (v132) (27), chimeras were removed using VSearch (28), and taxonomy was assigned with the RDP database (v18) (29). Non-bacterial sequences were removed, and remaining sequences were clustered at 97% similarity for downstream analysis.

All samples were rarefied to 3,047 reads which excluded two stool samples, four saliva samples, and all but one kit negative (of 18 kit negatives in total). The negative control remaining after rarefaction was likely due to cross-contamination during processing, indicated by its similarity to stool samples (all stool samples versus surviving negative control based on weighted UniFrac distances permutational multivariate analysis of variance (PERMANOVA): $R^2 = 0.00129$, $P$ value = 0.799). In total, 371 stool and 402 saliva samples were retained for analysis.

## Extremely preterm infant data set

To investigate the impact of gestational maturity on early stool and saliva microbes, a pre-existing 16S rRNA gene sequencing data set from previously published work focusing on the most extremely preterm infants (EP; born <28 weeks of gestation; $n = 7$) over the first 60 DOL was included for analysis (15). EP infants were born between November 2014 and November 2015 at the same hospital as LMPT infants. All EP infants received antifungals and probiotics (liquid preparation of *Lactobacillus acidophilus*, *Bifidobacterium bifidum*, and *B. infantis* (Labinic, Biofloratech, UK) from the date of first full feed, were ventilated from birth, and cared for in the neonatal intensive care unit (NICU). Of the seven EP infants, two had sepsis (culture positive), one had necrotizing enterocolitis (NEC) (medical), and one had both sepsis (culture positive) and NEC (surgical). The remaining three EP infants had no diagnosed disease, including sepsis or NEC. An identical extraction kit was used for stool; however, saliva samples were collected by suction and extracted using PowerFood DNA Microbial Isolation Kits (QIAGEN). Sequencing was conducted using an identical protocol, and analyses were conducted at the same rarefaction level of 3,047 reads. All LMPT samples obtained over the first 60 DOL were included for comparison and grouped according to gestational age (GA) into moderate (MP; 32 to 34 weeks of gestation, $n = 33$) or late (LP; 34 to 36 weeks of gestation, $n = 121$) preterm groups. To address challenges of repeated measures during analysis and to facilitate the identification of potential changes in microbial composition over time, two distinct time windows were established (DOL 0–25 and DOL 26–60, largely equivalent to Entry and TEA time points in LMPTs, respectively).

## Statistical analysis

All statistical analyses were performed in R software version 4.3.1 (30) on rarefied samples. Unless stated otherwise, all visualizations were produced using the ggplot2 package (v3.4.3) (31). Excluding the analysis with the EP infant group, comparisons considered 10 groups that were established according to visit (Entry, TEA, 3, 6, or 12 months CA) and sample type (stool or saliva), to offer a comprehensive examination of spatiotemporal variations in microbial composition. $P$ values < 0.05 were deemed significant.

## Clinical data

Continuous variables are presented as median (interquartile range [IQR]), and categorical variables are shown as proportions. Differences between groups were calculated using Fisher's or $\chi^2$ test (dependent on cell counts) for categorical variables or Kruskal-Wallis for continuous variables.

## Alpha diversity

Alpha diversity based on number of OTUs (absolute count of species-level OTUs, representing richness) and Shannon diversity (richness and evenness) was computed for all samples using the vegan package (v2.6.4) (32). Groups were compared using Wilcoxon rank sum tests or linear regression models from the "rlm" function in package MASS (v7.3.60.0.1) (33). The models were adjusted for factors reported to be associated with infant microbiomes that were available for both LMPT and EP data sets; these included sample DOL, delivery mode, season of sample collection, postnatal antibiotics (any Y/N), naso-gastric feeding (ever Y/N), and feeding mode (breastmilk, formula, or mixed). Birth weight was excluded from the models as it was highly correlated (R > 0.8) with GA. Global *P* values for variables from the final fitted models were obtained by type II analysis of variance in the car package (v3.1.2) (34). *Post-hoc* analysis was performed by pairwise comparisons (Tukey's HSD method) from the emmeans package (v1.10.0) (35).

## Beta diversity

Beta diversity was observed based on weighted and unweighted UniFrac using the rbiom package (v1.0.3) (36). Median distances for within or between group comparisons were compared using Wilcoxon rank sum tests. PERMANOVA was performed with 1,000 permutations using the "adonis2" function of the vegan package (32). Ordinations were conducted using non-metric multidimensional scaling (NMDS) by the "metaMDS" function of the vegan package (32). Stress values were calculated to assess goodness-of-fit, and Shepard plots were generated to visually inspect the correspondence between original dissimilarities and distances in the NMDS solution. Ordinations were permissible with stress values < 0.15.

## Taxonomy

Taxonomic relative abundance of stool and saliva samples was explored at the phylum and genus level using default parameters in MaAsLin2 (v1.14.1), where q-values of <0.25 were deemed significant (37). Infant factors previously associated with the microbiome, including delivery mode, season of sample collection, feeding mode (breastmilk, formula, or mixed), postnatal antibiotics (any), GA, sex, siblings, naso-gastric feeding (ever), and birth weight, were incorporated as fixed effects in LMPT analyses, while the comparison with EP infants used the same fixed effects where the corresponding metadata was available. Unique patient identifier and twin/triplet group identifier were included as random effects. Each related set was assigned a single group identifier, while individuals retained their own unique identifiers. Wilcoxon rank sum tests were used to detect significant variations in phyla or genera between adjacent time points/windows in stool and saliva. Venn diagrams were plotted with the ggVennDiagram package (v1.5.0) (38).

## Relationship between gut and oral microbiomes

Alpha diversity metrics of paired stool and saliva samples were correlated using Kendall's tau at each time point. Procrustes analysis was conducted at each time point to evaluate congruency between calculated stool and saliva distance matrices, using the "procrustes" function from the vegan package, and significance was calculated using the "protest" function with 1,000 permutations, also from the vegan package (32). Procrustes residuals at Entry and TEA were regressed in univariable generalized linear models with

sample age, GA, sex, siblings, season of sample collection, delivery mode, feeding mode (breastmilk, formula, or mixed), postnatal antibiotics (any), naso-gastric feeding (ever), and birth weight. Paired weighted UniFrac distances were regressed using the same models to corroborate findings. Genera driving separation between sample types was investigated in paired samples by linear discriminant analysis effect size (LEfSe) at each time point. LefSe was run with effect size threshold >2 using the lefser package (v 1.10.1) (39).

Where appropriate in all analyses, *P* values were adjusted for multiple comparisons with FDR correction (40).

## RESULTS

### LMPT cohort samples and characteristics

A total of 371 stool and 402 saliva samples from 160 LMPT infants were included for analysis. Owing to recruitment and sample collection challenges during the COVID-19 pandemic, many babies had some missing data, but most infants provided stool and/or saliva for the majority of time points. 145 (91%) contributed stool samples (median [IQR] of 3 [2, 3] samples) and 158 (99%) provided saliva samples (median [IQR] of 3 [1–3] samples). Less than 50% of enrolled infants contributed samples at 6 months CA and less than 20% at 12 months CA. Study infants had a median gestational age (GA) of 35 + 3 weeks (IQR: 34 + 2 to 36 + 2) and birth weight of 2.23 kg (IQR: 2.07 to 2.43). Despite incomplete sample collection, underlying patient characteristics did not vary significantly between sample types or time points (all *P* values > 0.05; Table S1).

### Development of LMPT infant gut and oral microbiomes over the first year of life

Stool and saliva both demonstrated increasing bacterial richness and Shannon diversity with age, increases that were statistically significant between each adjacent time point (all *P* values < 0.05), excluding Shannon diversity between TEA and 3 months CA (stool *P* = 0.167; saliva *P* = 0.248) (Fig. 1). The largest increases in alpha diversity were between 6 and 12 months CA in the stool, and between 3 and 6 months CA in the saliva. Comparing the alpha diversity of sample types within each time point showed that stool samples were significantly more rich and more diverse than saliva samples at all time points; this difference was most pronounced at 12 months CA (all *P* values < 0.05; Fig. 1).

The stool microbiome profiles showed significant increases in similarity among individuals from entry up to 3 months CA, after which individuals became significantly more dissimilar at each successive time point (*P* values < 0.001; Fig. S1a). In contrast to stool, saliva samples became more similar after 3 months and were most similar at 12 months CA. Despite entry to TEA representing the shortest period between time points, stool showed the greatest change to microbiome profiles during this period (Fig. S1b). NMDS ordination provided further evidence of spatiotemporal variability, revealing sample type-specific clustering that became more pronounced over time (all *P* values < 0.001; Fig. 2). The influence of sample type on microbial composition was substantial, explaining 27% of the variance at entry and 48% of the variance at 12 months CA (Fig. 2), again emphasizing the increasing dissimilarity between sample types over time.

Over the entire first year of life, LMPT stool samples exhibited dynamic changes in the relative abundances of the four dominant phyla (Bacillota, Pseudomonadota, Actinomycetota, and Bacteroidota), with all but Bacillota showing significant temporal variation after correcting for co-variates (*q* values < 0.001; Fig. 3a; Table S2). At the entry time point, stool primarily consisted of Pseudomonadota (average relative abundance 50%, detected in 97% of samples) and Bacillota (41% in 98% of samples). Actinomycetota relative abundance increased significantly up to 3 months CA and dominated the stool microbiome (mean relative abundance 47%; *P* values < 0.001). After 3 months CA, Bacillota relative abundance increased, becoming the highest at 12 months CA compared with all other time points (all *P* values < 0.01). At the genus level in LMPT stool

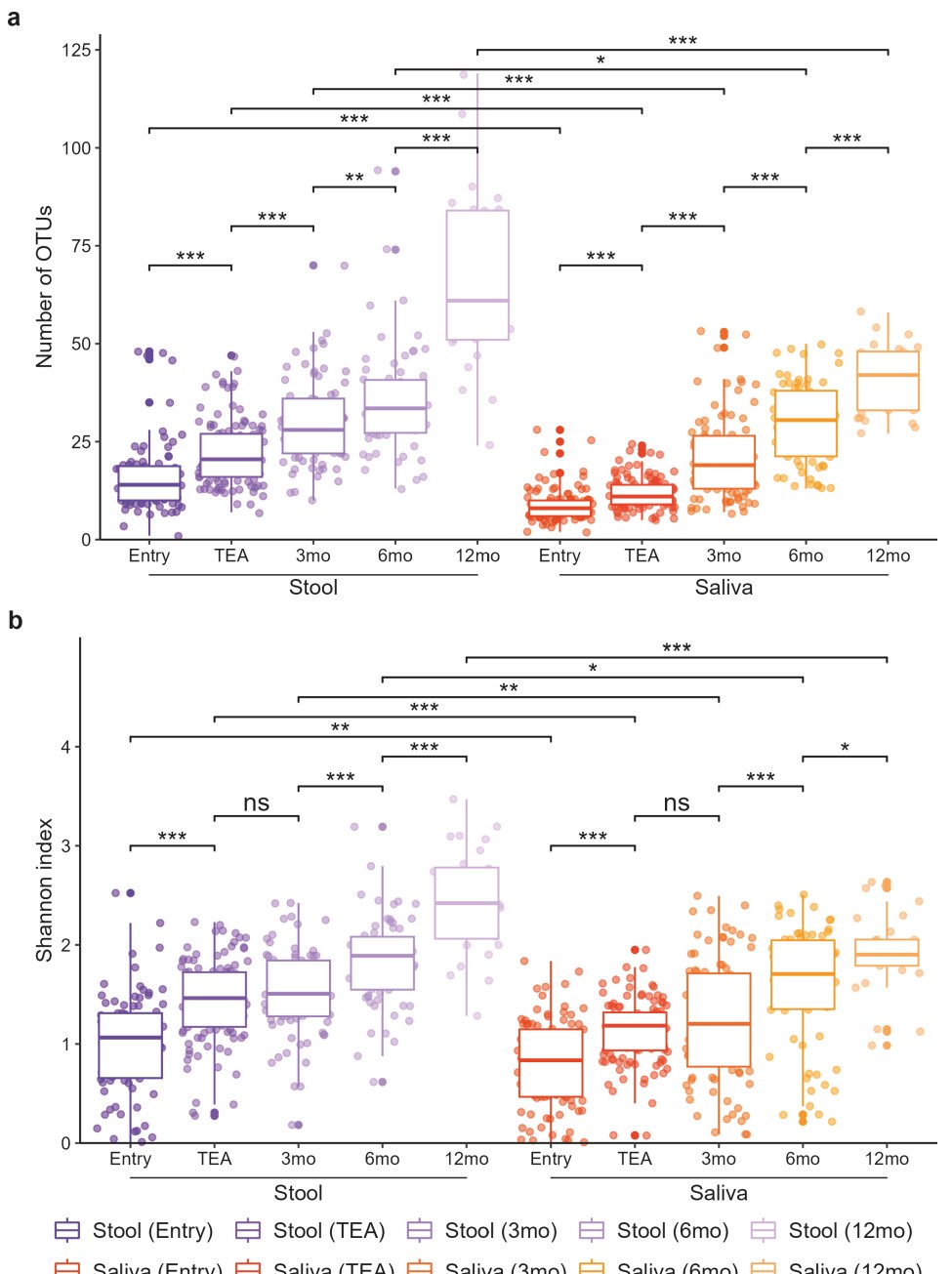

**FIG 1** Alpha diversity of late and moderate preterm stool and saliva samples over the first year of life. Plots show (a) the number of OTUs and (b) Shannon index of stool and saliva samples at given time points. Boxes are colored according to time point and sample type and show median (center line) and interquartile ranges (IQR) (box limits); whiskers extend ± 1.5*IQR from box's quartile. Points outside the whiskers represent outliers. *P* values are from FDR adjusted Wilcoxon rank sum tests; ns indicates non-significant *P* value, * indicates *P* value < 0.05, ** indicates *P* value < 0.01, *** indicates *P* value < 0.001. TEA, term equivalent age; mo, months corrected age.

samples, five taxa persisted in abundance (>1%) across all time points: *Bifidobacterium*, *Enterococcus*, *Escherichia/Shigella*, *Streptococcus*, and *Veillonella* (Fig. 3b). Pseudomonadota were largely represented by members of the Enterobacteriaceae family (*Escherichia/Shigella* and *Klebsiella*) across all time points (Fig. 3b). Within each time point, the top five most abundant genera from Bacillota ranged from 3% to 13% in mean relative abundance and included *Streptococcus, Veillonella, Staphylococcus, Enterococcus, Clostridium sensu stricto, Lacticaseibacillus, Blautia, Faecalibacterium,* and unclassified genera from the

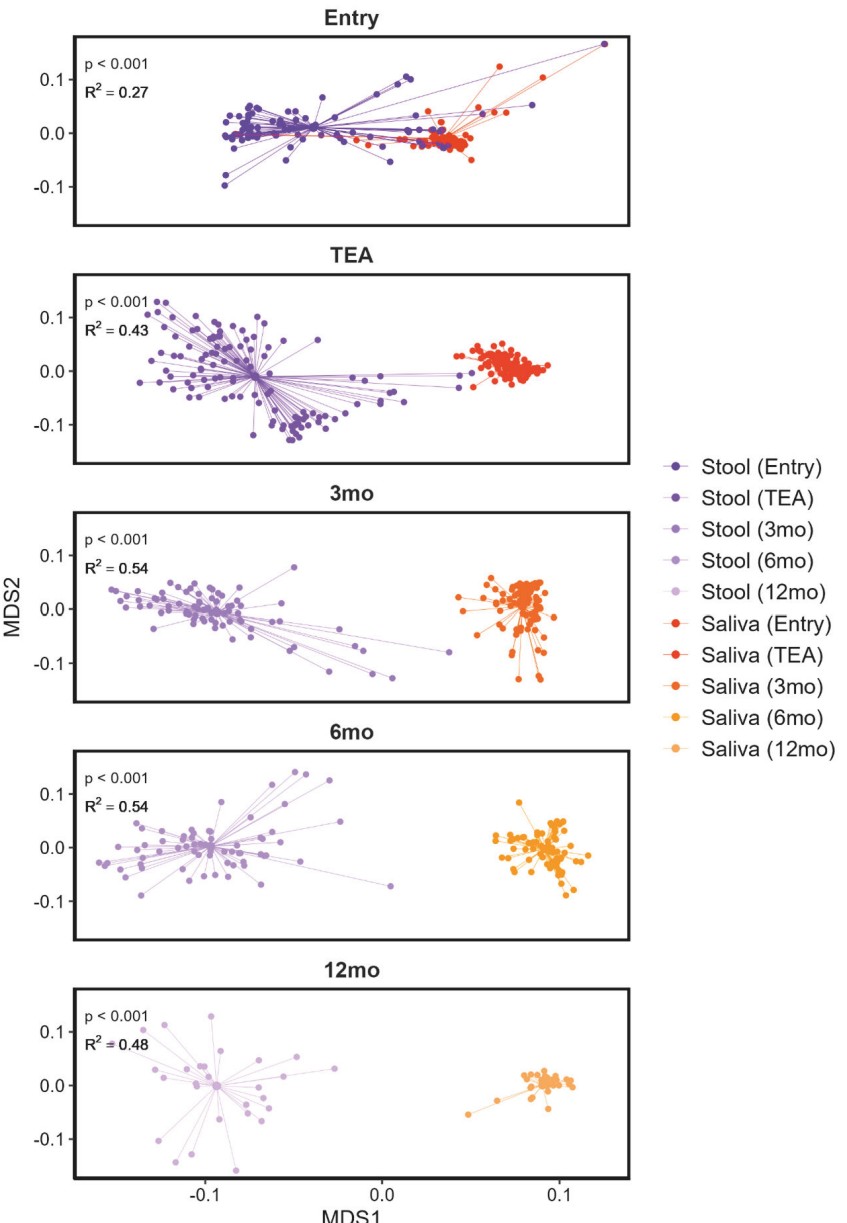

**FIG 2** NMDS plots of late and moderate preterm stool and saliva samples over the first year of life. Plots are based on ordination of weighted UniFrac distances and show the mean centroids for each group. Each point represents an individual sample. $P$ values and $R^2$ values on plot faces are from PERMANOVA. TEA, term equivalent age; mo, months corrected age.

families Lachnospiraceae and Ruminococcaceae, many of which showed significant changes between adjacent time points ($P$ values < 0.05; Fig. 3b). Actinomycetota were largely dominated by *Bifidobacterium* (at least 84% of total Actinomycetota) across all time points. *Bifidobacterium* relative abundance varied significantly over the first year of life ($q$ < 0.001; Table S2) reaching peak abundance at 3 months CA (Fig. 3b).

In comparison to stool, the LMPT salivary microbiome showed greater homogeneity, with remaining dominated by Bacillota phylum over the entire observed period (mean relative abundance 84%, prevalence was 100% at all time points; Fig. 3a). All four dominant phyla showed significant temporal variation in their relative abundances (q

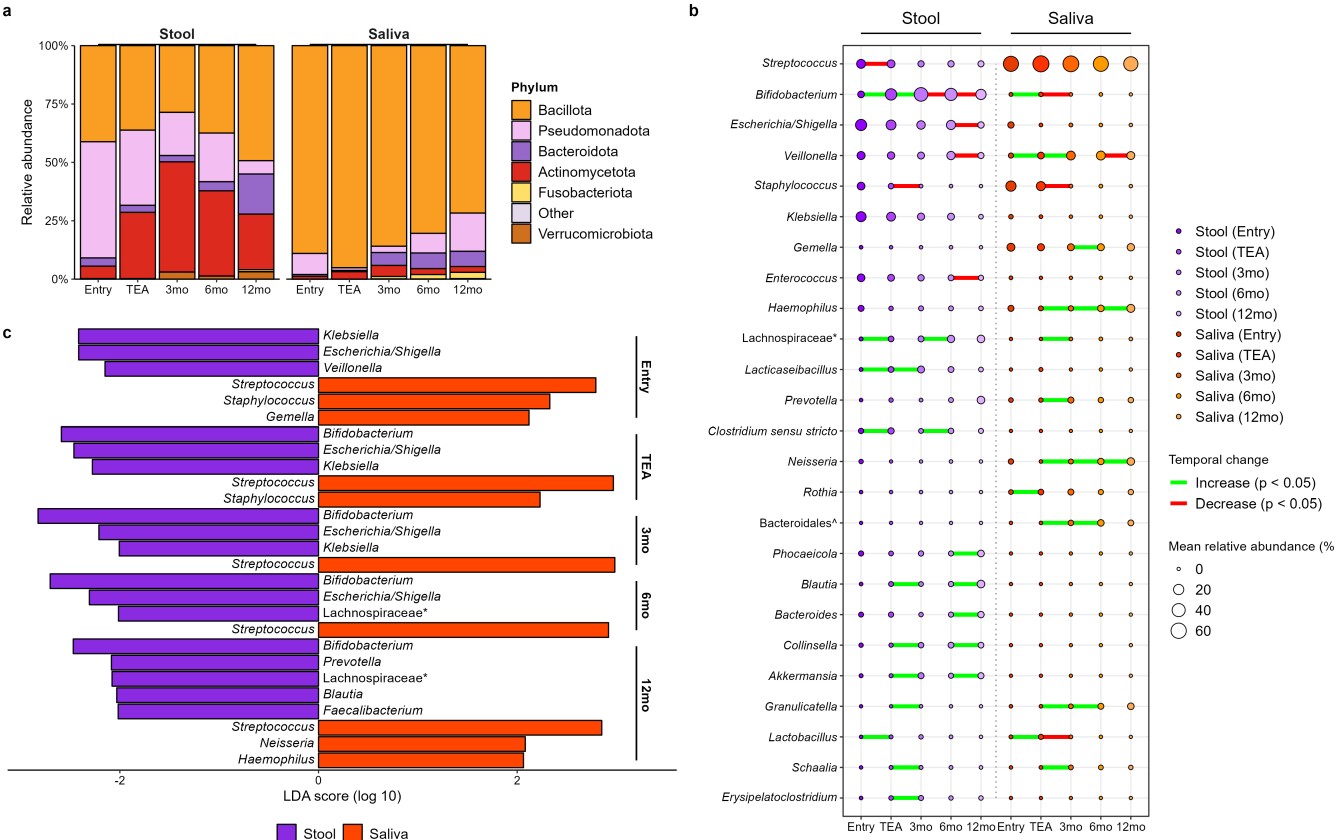

**FIG 3** Overview of the taxonomic compositions of late and moderate preterm stool and saliva samples over the first year of life. (a) Stacked bar charts showing the average relative abundances of bacterial phyla detected in stool and saliva samples at given time points. (b) The top 25 most abundant genera found in all samples separated by sample type and time point. Bubble size represents average relative abundance of each genus within the group. Connecting lines show significant (FDR *P* value < 0.05) increases (green) or decreases (red) in relative abundance between consecutive time points. (c) LEfSe analysis of genera significantly discriminating paired stool and saliva samples at each time point. The plot is based on linear discriminant analysis (LDA); genera with LDA score > 2 and *P* value < 0.05 are shown. Bars are colored according to sample type. Unclassified genera are represented by their family (*) or order (^). TEA, term equivalent age; mo, months corrected age.

values < 0.25; Table S3). Although Bacillota remained prominent in saliva, its composition varied over time, with many of the genera comprising the phylum showing significant variation across time and between adjacent time points (*P* values < 0.05; Fig. 3b; Table S3). *Streptococcus* accounted for 66%–77% of Bacillota at each time point; the remainder of Bacillota generally consisted of *Staphylococcus* at earlier stages and *Veillonella* later. A complete list of genera that showed significant temporal variation is provided in Tables S2 and S3.

## Relationship between paired LMPT stool and saliva microbiomes

To explore potential relationships between stool and salivary microbiomes, a subgroup of infants that provided both sample types concurrently at similar timepoint(s) was explored. A total of 674 paired samples (337 each of stool and saliva) were collected at nearly identical ages (average difference = 0.55 days; Spearman's *R* = 1, *P* < 0.001; Fig. S2). Correlations of alpha diversity from paired stool and saliva samples were mostly non-significant (Fig. S3). Notably, although alpha diversity of stool was greater than saliva at the cohort level, paired samples showed that some infants had higher richness (63/337, 19%) and Shannon diversity (111/337, 33%) in their saliva samples (Fig. S3).

Procrustes analysis based on weighted UniFrac distances revealed significant correspondence between individual's stool and saliva at entry (*R* = 0.331, *P* = 0.005)

and TEA ($R = 0.252$, $P = 0.013$) (Fig. S4a). At later time points (3, 6, and 12 months CA), correspondence was not significant ($P > 0.05$), which may result from both the temporal diversification of stool samples and the reduced number of sample pairings, limiting statistical power. Post-hoc power analyses indicated less than 30% power at all three later time points. Univariable regression of the resulting residuals found that feeding mode (breastmilk, formula, or mixed) was significant in predicting correlations between stool and saliva samples at TEA ($P$ values < 0.05; Fig. S4b). The microbiome profile of paired stool and saliva samples was most similar in formula-fed infants, while infants receiving breastmilk were least similar. Regression of paired weighted UniFrac distances corroborated this finding ($P = 0.027$; Fig. S4c).

LEfSe analysis within each time point showed the discriminatory genera between sample types changed over time as the microbiome diversified, with the exception of *Streptococcus* which discriminated saliva samples across all time points (Fig. 3c). *Streptococcus* showed that the highest effect sizes (largest LDA score) overall and was the most discriminatory genera in saliva samples at all time points, despite being consistently present in both sample types (Fig. S5). In stool samples, *Klebsiella* had the largest discriminatory power at entry, while *Bifidobacterium* consistently displayed the largest effect sizes thereafter (Fig. 3c).

## Comparison of early LMPT stool and saliva microbiome development to extremely premature infants

This analysis comprised 212 stool (14 extremely preterm (EP), 48 moderate preterm (MP), and 150 late preterm (LP)) and 222 saliva (14 EP, 49 MP, and 159 LP) samples (Fig. S6). Differences in characteristics between GA groups primarily reflected expected differences in gestational maturity and accompanying clinical care; for instance, birth weight increased proportionally with increasing GA (Spearman's $R = 0.82$, $P < 0.001$), while length of hospital stay was inversely related to GA (Spearman's $R = -0.94$, $P < 0.001$) (Tables S4 and S5). All infants in the EP group were treated with antibiotics and in the hospital at the time of sample collection. EP infants in this analysis were more likely to have been delivered vaginally (significant during DOL 26–60, $P = 0.025$), and all EP infants in this group received enteral feeding exclusively with expressed maternal breastmilk delivered via naso-gastric tubes. In contrast, MP and LP infants were fed breast milk either directly or via naso-gastric tubes, received formula, or a combination of both (mixed feeding). All EP infants were supplemented with probiotics, whereas MP and LP infants were not. Due to limited availability of samples, the variation in sample DOL among the groups was significant ($P$ values < 0.01; Fig. S6).

Over the first 60 DOL, alpha diversity of stool and saliva samples increased significantly for MP and LP infants (all $P$ values < 0.01) but not EP infants (all $P$ values > 0.05) (Fig. S7). To account for potential confounding variables, alpha diversity for each GA group was then estimated using linear regression models (see Methods), showing stool alpha diversity from EP infants was lower than that of MP and LP infants, whereas saliva samples from EP infants had higher alpha diversity than both MP and LP infants (Fig. S8). The differences in stool alpha diversity were most evident within the first time window (DOL 0-25; bacterial richness $P < 0.001$ and Shannon diversity $P = 0.071$). Conversely, the greater alpha diversity in EP saliva samples became more significant with time. Weighted UniFrac distance matrices showed significant separation between GA groups and sample types at both time windows (global PERMNOVA $P$ values < 0.05; Fig. 4a). For stool microbiome, EP infants were significantly different from MP and LP infants only at the later time window ($P = 0.007$ and $0.01$, respectively). In saliva, EP infants lacked divergence of the bacterial community from that of stool (all $P$ values > 0.05).

Pseudomonadota and Bacillota were the dominant phyla in stool across all GAs at DOL 0–25, but in the later time window (DOL 26–60) Pseudomonadota dominated in EP infants ($q$ values < 0.001), and Bacillota was the most dominant phyla in MP and LP infants (Fig. S9a and b). In saliva samples, the relative abundances of Bacteroidota and

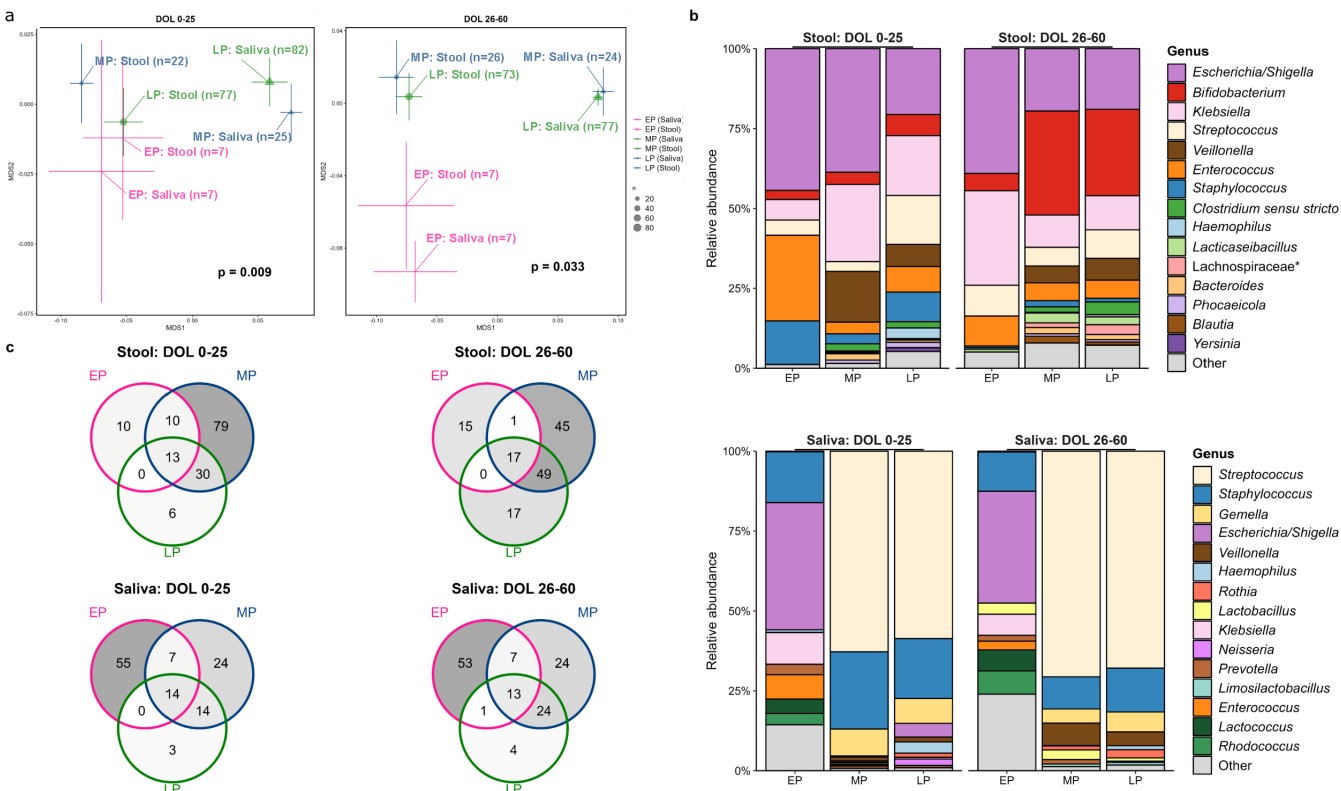

**FIG 4** Stool and saliva microbiome across all preterm gestational age groups at given time windows. (a) NMDS plots of stool and saliva samples based on gestational age at given time windows. NMDS plots show clustering by gestational age group and sample type at DOL 0–25 and DOL 26–60. Plots are based on ordination of weighted UniFrac distances and show the mean centroids for each group. Centroid size is based on the number ($n$) of samples and bars represent the ±95% CI. $P$ values and $R^2$ values on plot faces are from global PERMANOVA. (b) Stacked bar charts show the average relative abundances of the top 15 most abundant genera found in stool and saliva samples in extremely, moderate, and late preterm infants. (c) Shared and unique genera in stool and saliva based on gestational age at given time windows. Venn diagrams show the number of shared and unique genera detected in stool and saliva samples between DOL 0–25 and DOL 26–60. Plots are colored according to gestational age group. Unclassified genera are represented by their family (*). DOL, day of life; EP, extremely preterm; MP, moderate preterm; LP, late preterm.

Pseudomonadota were significantly lower, and Bacillota was significantly higher in MP and LP compared with EP infants at both time windows ($q$ values < 0.05; Fig. S9c and d).

At the genus level, MP and LP stool, but not EP stool, showed significant changes in the relative abundances of stool genera over time ($P$ values < 0.05). During the initial time window (DOL 0–25), *Escherichia/Shigella* dominated stool in all GA groups, decreasing in relative abundance and prevalence with increasing GA (Fig. 4b). Focusing on differences in stool bacterial genera between EP and more mature MP and LP infants showed that *Enterococcus* was significantly higher in EP infants, and *Veillonella* was significantly higher in more mature infants between DOL 0 and 25 ($q$ values < 0.25; Fig. 4B; Table S6). In the saliva between DOL 0 and 25, both MP and LP infants were dominated in order of rank abundance by *Streptococcus, Staphylococcus,* and *Gemella* (Fig. 4B). EP infant saliva harbored significantly higher proportions of *Escherichia/Shigella*, *Klebsiella,* and *Enterococcus* ($q$ values < 0.001), while *Streptococcus* and *Gemella* were enriched in more mature infants (q values < 0.05; Fig. 4B; Table S7). Similar to EP stool, EP saliva remained relatively stable over time, remaining dominated by *Escherichia/Shigella* in comparison to more mature groups ($q$ values < 0.001; ; Table S7). During the second time window (DOL 26–60), MP and LP infant saliva also had significantly higher abundances of *Rothia* and *Veillonella* ($q$ values < 0.25; Fig. 4B; Table S7). MP infants had the greatest number of unique stool genera across both time windows, which may partly reflect the increased DOL within this group (Fig. 4c).

## DISCUSSION

To the best of our knowledge, the current study is the first to comprehensively describe concurrent gut and oral microbiota development in an understudied population of LMPT infants. In addition, we compared these LMPT infants with a group of EP infants cared for in the same hospital. Together with the differences between temporal development of microbiomes in stool and saliva of LMPT and EP infants, our results highlight the unique trajectories of LMPT gut and oral microbiomes and further underscore the importance of gestational maturity in shaping microbial communities.

In this LMPT cohort, the earliest most dominant colonizers of stool were taxa from Family Enterobacteriaceae (*Escherichia/Shigella* and *Klebsiella*), which are recognized, often abundant members of extremely preterm/low birth weight infant stool (41, 42), and have been reported to dominate early LMPT stool in other studies (20, 21). Although it represented the shortest duration of time, the largest changes in stool microbiota composition were observed between entry and TEA. The gut microbiota rapidly evolved as the gut is known to become more anaerobic, introducing several anaerobic bacteria from Order Clostridiales, *Lacticaseibacillus*, and *Lactobacillus,* among others, along with increasing abundances of *Bifidobacterium*, which became the predominant genus in LMPT stool from TEA onwards. This contrasts one previous report showing delayed *Bifidobacterium* dominance in probiotic-naïve LMPT infants until 6 months (21); however, the lack of a time point between 1 and 6 months in the study by reference 21 may account for this discrepancy. The finding is generally in line with another report that *Bifidobacterium* dominates the LMPT infant gut microbiome from 28 days (20). We saw a period of accelerated convergence among infants concurrent with a *Bifidobacterium* bloom, which has also been recorded in full-term infants at 2–4 months of age, the end of this bloom coinciding with the introduction of solid food (43). Indeed, the majority of LMPT infants in this study were receiving solids by 6 months CA, explaining the reduction in *Bifidobacterium* and interindividual deviation seen in stool after 3 months CA. Increasing microbial diversity and declining Pseudomonadota from birth, coupled to expansion of Bacteroidota over time, were generally in line with previous LMPT studies (18, 20). At 12 months, LMPT infant stool was dominated by *Bifidobacterium* and taxa from Order Clostridiales, including Lachnospiraceae, *Blautia*, *Faecalibacterium,* Ruminococcaceae, *Anaerostipes, Mediterraneibacter*, and *Clostridium sensu stricto*. Many of these taxa became abundant or prevalent only at this later time point, reflecting a more mature profile like that observed in full-term infants at the same stage (44, 45).

LMPT saliva was dominated by *Streptococcus* across all timepoints, which is comparable to that of term infants (46, 47). LMPT oral microbiome research is limited; however (16), reported that early moderate preterm (MP) samples were initially dominated by *Staphylococcus*, with increasing abundances of *Streptococcus* and *Rothia* as the infants aged and transitioned from tube to direct breastfeeding (16, 22). Observed late preterm (LP) infants during the first week of life and similarly reported increasing abundances of *Streptococcus* and *Rothia* with age, although they did not specify which taxa dominated overall (22). In our study, the LMPT cohort included a larger proportion of LP infants (78%) who were more likely to be directly breastfed, which may explain the initial dominance of *Streptococcus* observed.

This LMPT cohort presented with distinct microbial profiles from the earliest sampling point, with sample type explaining more of the variance than time point. Previous studies have reported distinct early infant gut and oral microbiomes (22, 48–51), while others have reported a lack of differentiation between body sites in preterm and even full-term infants shortly after birth (22, 41, 52). Despite representing separate body sites with distinct microbial niches, the oral cavity and the gut are physiologically linked via the gastrointestinal tract, and evidence suggests that microbes may be seeded from oral cavity to gut, even in healthy adults (53). Schmidt and colleagues found sharing of strains from *Streptococcus, Veillonella, Actinomyces, Haemophilus, Prevotella, Fusobacterium, Gemella, Rothia, Bifidobacterium, Lactobacillus,* and *Granulicatella* (53), all of which were part of the top 10 most shared genera in paired stool and saliva samples in this

cohort. Although Ferretti et al. only explored infants up to day 3 of life, strains from all top genera shared at entry and TEA in this cohort, except *Klebsiella,* were found to be transmissible from infant oral cavity to gut (50). It was of interest to find that the similarity between paired stool and saliva samples at TEA was related to feeding mode, with formula encouraging colonization in both sample types by similar abundant taxa. While the present results point towards evidence of microbial seeding in LMPT infants, further investigation at the strain level is needed. Additionally, we cannot exclude the possibility that the increased dissimilarity in the stool and saliva of infants receiving breast milk may be partly explained by the bifidogenic effect of breastmilk, which, due to its high content of prebiotic human milk oligosaccharides, may promote the growth of *Bifidobacterium* and other beneficial microbiota (54).

The LMPT cohort followed microbiome development patterns as previously seen in full-term infants, characterized by increasing alpha diversity, dynamic compositional shifts, and an ordered succession of specific taxa colonizing the gut and mouth (44, 47, 55–57). While mature MP and LP groups showed development of both gut and oral ecosystems in the first 60 DOL, the taxonomic compositions of EP infants' stool and saliva remained relatively stable. Still, the EP infant group may have been underpowered to detect differences (19). Forsgren et al. reported that GA modulated the gut microbiome independently of other known risk factors (intrapartum and postnatal antibiotics, delivery mode, and breastfeeding) (19). Notwithstanding, EP infants are hospitalized and exposed to more microbiome modulating therapies (e.g., antibiotics and probiotics), so distinction of EP from LMPT infants was not unexpected. This suggests hospitalization over prematurity as the driver of the infant microbiome, but would require a hospitalized LMPT group to confirm. *Bifidobacterium* colonization in stool has been previously associated with GA (17, 20) and is typically low abundance in EPs not receiving probiotics (10). Notably, we found no statistically significant differences in *Bifidobacterium* between EP and MP/LP infants over the first 60 days. This likely reflects the universal use of *Bifidobacterium*-containing probiotics to EP infants in the NICU, while there was no probiotic use for MP/LP infants. Saliva samples exhibited an increased impact of GA; EP infants became increasingly dissimilar from LMPT groups over time, despite EPs lacking any notable temporal differentiation. Early preterm oral samples have previously been reported to have higher alpha diversity than those from full-term infants (5), and we extend this to show this difference even among subgroups of preterm infants. Selway and colleagues additionally reported that preterm infant oral communities resembled full-term infants by 36 weeks postmenstrual age (5); this may explain the lack of *Streptococcus* domination in EP infants who only reached a maximum of 31 completed weeks postmenstrual age within this study, contrasting the MP and LP infants.

The study suffered attrition over the course of the sampling period due to the COVID-19 pandemic, meaning that longitudinal analysis had to be conducted on a cohort/population level rather than individual. This may have concealed specific temporal trends at the individual level. While mixed-effects models were used to account for repeated measures in overall temporal taxonomic analyses, pairwise group-level comparisons between adjacent time points were conducted without modeling repeated measures. This approach may have introduced partial dependence between observations and potentially inflated Type I error rates. In addition, all LMPT samples were collected from infants born in a single hospital and represented a homogenous population that primarily reflected location, with the majority of mothers self-reporting as White British, meaning that findings may not be generalizable to other geographical locations. Indeed, a significant impact of site on preterm microbial communities has been previously reported (58). Analysis may also have been limited by the use of 16S rRNA gene sequencing resulting in a lack of resolution to strain level, but given saliva has high host DNA and relatively low microbial biomass, it was decided an amplicon-based approach would be most robust. The V4 region was selected for consistency with most previous work, including the EP cohort, but other variable regions may offer improved resolution for some oral taxa. Moreover, differences in extracting DNA and methods of

sample collection (EP oral samples collected by suction rather than swab) may have contributed to some of the differences observed between EP and LMPT infants. Finally, the limited size and unique clinical course of the EP cohort may confound this analysis, but nonetheless reflects the real-world scenario for EP compared to the LMPT infants.

In summary, this study provides a comprehensive description of gut and oral microbiome development in otherwise understudied LMPT infants over the first year of life. In LMPTs, both body sites were compositionally distinct and showed significant temporal maturation, with developmental patterns similar to those reported for full-term infants, albeit with delayed acquisition of certain taxa. The study additionally highlighted the significant impact of GA, namely, hospitalization and NICU clinical practice, on early microbial communities. The unique differential microbial trajectories observed between EP and LMPT infants underscore the need for tailored interventions to support optimal microbial development in subgroups of preterm infants. Further research is needed to investigate how delays in the acquisition of specific taxa impact later health and development in LMPT infants.

## ACKNOWLEDGMENTS

This project was supported by an unrestricted research grant from Danone Early Life Nutrition paid to Newcastle Hospitals NHS Foundation Trust. The funders had no role in study contact, data acquisition, or analysis. S.A.-F. is funded by the MRC Discovery Medicine North Doctoral Training Partnership.

## AUTHOR AFFILIATIONS

[1]Translational and Clinical Research Institute, Newcastle University, Newcastle upon Tyne, United Kingdom

[2]Newcastle Neonatal Service, Royal Victoria Infirmary, Newcastle upon Tyne Hospitals National Health Service (NHS) Foundation Trust, Newcastle upon Tyne, United Kingdom

[3]Department of Applied Science, Northumbria University, Newcastle upon Tyne, United Kingdom

[4]Danone Nutricia Research, Utrecht, the Netherlands

[5]Emma Children's Hospital, Amsterdam University Medical Centers, Amsterdam, the Netherlands

[6]Hub for Biotechnology in the Built Environment, Northumbria University, Newcastle upon Tyne, United Kingdom

[7]Population Health Sciences Institute, Newcastle University, Newcastle upon Tyne, United Kingdom

## AUTHOR ORCIDs

Darren Smith http://orcid.org/0000-0003-4925-467X
Christopher J. Stewart http://orcid.org/0000-0002-6033-338X

## FUNDING

| Funder | Grant(s) | Author(s) |
| --- | --- | --- |
| Danone Early Life Nutrition | NA | Janet E. Berrington |
| | | Nicholas D. Embleton |

## AUTHOR CONTRIBUTIONS

Sinéad Ahearn-Ford, Conceptualization, Data curation, Formal analysis, Funding acquisition, Supervision, Writing – original draft | Andreas Kakaroukas, Conceptualization, Data curation, Formal analysis, Project administration, Writing – original draft, Writing – review and editing | Gregory R. Young, Conceptualization, Data curation,

Formal analysis, Project administration, Writing – review and editing | Andrew Nelson, Data curation, Formal analysis, Resources, Writing – review and editing | Marieke Abrahamse-Berkeveld, Conceptualization, Data curation, Resources, Writing – review and editing | Ruurd M. van Elburg, Conceptualization, Writing – review and editing | Darren Smith, Conceptualization, Data curation, Writing – review and editing | Janet E. Berrington, Conceptualization, Data curation, Supervision, Writing – review and editing | Nicholas D. Embleton, Conceptualization, Funding acquisition, Supervision, Writing – review and editing | Christopher J. Stewart, Conceptualization, Funding acquisition, Supervision, Writing – original draft, Writing – review and editing

## DATA AVAILABILITY

The data that support the findings of this study are openly available in the NCBI BioProject database at http://www.ncbi.nlm.nih.gov/bioproject/1200902, reference number PRJNA1200902.

## ETHICS APPROVAL

The study was approved by the York Research Ethics Committee (ref: 18/NE/0040) and registered with the NHS Health Research Authority (IRAS project ID: 237542). Written parental/guardian consent was obtained before enrolmentenrollment into the study.

## ADDITIONAL FILES

The following material is available online.

### Supplemental Material

**Supplemental figures (mSystems00667-25-s0001.pdf).** Figures S1 to S9.
**Supplemental tables (mSystems00667-25-s0002.xlsx).** Tables S1 to S7.

### Open Peer Review

**PEER REVIEW HISTORY (review-history.pdf).** An accounting of the reviewer comments and feedback.

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
