## [Reviewer comments · mSystems]

Spatiotemporal development of late and moderate preterm infant gut and oral microbiomes and impact of gestational age on early colonisation

Sinead Ahearn-Ford, Andreas Kakaroukas, Greg Young, Andrew Nelson, Marieke Abrahamse-Berkeveld, Ruurd van Elburg, Darren Smith, Janet Berrington, Nicholas Embleton, and Christopher Stewart

Corresponding Author(s): Christopher Stewart, Newcastle University

Review Timeline:

Submission Date:	May 8, 2025
Editorial Decision:	July 4, 2025
Revision Received:	October 16, 2025
Accepted:	October 21, 2025

Editor: David Cleary

Reviewer(s): The reviewers have opted to remain anonymous.

Transaction Report:

DOI: <https://doi.org/10.1128/msystems.00667-25>

Re: mSystems00667-25 (Spatiotemporal development of late and moderate preterm infant gut and oral microbiomes and impact of gestational age on early colonisation)

Dear Prof. Christopher J Stewart:

Thank you for the privilege of reviewing your work. Below you will find instructions from the mSystems editorial office and the reviewer comments.

Revision Guidelines

Sincerely,
David Cleary
Editor
mSystems

Reviewer #1 (Comments for the Author):

This paper describes the development in late and moderate preterm infant gut and oral microbiomes that 160 infants - 371 stool and 402 saliva samples. The data is valuable as in giving insights into the gut and oral microbiome development in late and moderate preterm infants, which is understudied according to the authors. This paper is highly descriptive which is not surprising due to the nature of the data, purely based on describing the community structure in these infants in both gut and oral microbiomes. The majority of the outcomes go in line with other preterm infant studies, concluding that the gut microbiome of late and moderate preterm infants are more similar to term infant, highlighting the role of gestational maturity in shaping

microbial communities.

Minor comments:

1. Figure 4a, probably better to label the late, moderate and extremely preterm infants in the figure for clarity as well as the n - number of samples.
2. For clarity to general readers, perhaps beneficial to define Entry, Term Equivalent Age and corrected gestational age in line 47-48 (page 3).
3. Is Figure 1a referring to bacterial richness index Chao1 or absolute species/genus count? It says number of OTUs to me it means absolute species count. Also is data in Figure 1 based on genus or species?
4. Also in Figure 1, could the authors justify why Wilcoxon test was used rather than Kruskal Wallis?
5. In lines 10-13 (page 7), it says "all but Bacillota showing significant temporal variation...", this is at phylum level, however there is no single Genus that shows temporal variation between groups? Interpreting microbiome data at genus level will provide more meaningful insights to the readers. Otherwise consider to add a sentence to clarify Bacillota on a genus level.
6. It will be advisable to show relative abundance on genus level in Figure 3 alongside the phylum.
7. In Figure 2, it may be better to show also the overall NMDS of all the samples, perhaps will visualise better the clustering of the samples according to the microbiome data?

Major comments:

1. Authors should attempt to simplify the results and discussion section and focus only on the key findings and data.
2. The authors should also make it clear in the beginning that this cohort of infants are probiotic-supplemented (to what I understood) as this impacts the data interpretation and conclusion.

Reviewer #2 (Comments for the Author):

This manuscript by Ahearn-Ford et al. presents a longitudinal analysis of stool and saliva microbiomes from 160 late and moderate preterm (LMPT) infants over the first year of life. The study identifies distinct and maturing microbial communities with increasing diversity over time. Given the relative paucity of data on LMPT infant microbiomes, this work provides a valuable contribution to the field. The manuscript is clearly written and well-organized, with extensive supplementary data. However, several methodological issues require clarification, and some limitations should be more addressed.

1. Clarification of Sample Size and Longitudinal Sampling: Although the authors highlight the cohort size (n=160) as a strength, it is important to clarify that no more than 115 samples were collected at any individual time point. A table summarizing the number and percentage of samples available at each time point should be included to improve transparency. Additionally, please specify how many infants contributed longitudinal samples (i.e., samples from more than one time point), as this is critical for interpreting the longitudinal analyses.
2. Choice of 16S rRNA Region: The rationale for selecting the V4 region for 16S rRNA sequencing over other commonly used regions (e.g., V1-V2 or V3-V4) is not provided. Please justify this choice, particularly in the context of the oral microbiome, where other variable regions may offer better resolution for specific taxa.
3. Longitudinal Analysis and Statistical Modeling: The division of the dataset into 10 groups to address repeated measures assumes independence of observations, which may not be appropriate for longitudinal data. This may result in inflated type I error rates and biased conclusions. The use of mixed-effects models would be more statistically rigorous and better account for within-subject correlations over time.
4. Beta Diversity and Pseudoreplication: Beta diversity was assessed using UniFrac distances and PERMANOVA (via the `adonis2` function). However, it is unclear whether permutations were stratified by patient ID. This is essential to avoid pseudoreplication in repeated measures designs. Please clarify this aspect of the analysis. Furthermore, indicate whether NMDS ordinations were performed after adjusting for within-subject correlation.
5. Oral-Gut Microbiome Correlation: Kendall's tau was used to assess correlation between oral and gut microbiomes at each time point. While this is informative, the authors may wish to consider additional approaches such as source-tracking or source-sink analysis to better characterize potential microbial transmission between body sites.
6. Inclusion of Extremely Preterm (EP) Cohort: The comparison with previously collected microbiome data from seven extremely preterm (EP) infants is limited by small sample size and significant differences in sampling time points, feeding practices, and antibiotic exposure. These limitations substantially reduce the interpretability of the comparison. It may be more appropriate to present this analysis in the supplementary material, along with a clear discussion of the caveats. This would also allow the main figures to focus on the LMPT cohort, which is the core strength of the study. Additionally, please note that the description of Figure 4c is missing from the main text.
7. Minor Comments: In the Introduction (lines 17-18), please cite relevant literature on the gut microbiome in extremely preterm infants. In the Methods section, clinical data is provided for only 4 of the 7 EP infants. Please include this information for all subjects. For clarity, indicate the number of samples represented in each panel or figure legend of Supplementary Figure 4.

We thank the reviewers for their comments. We have responded point-by-point to all reviewer comments below.

Reviewer #1 (Comments for the Author):

This paper describes the development in late and moderate preterm infant gut and oral microbiomes that 160 infants - 371 stool and 402 saliva samples. The data is valuable as in giving insights into the gut and oral microbiome development in late and moderate preterm infants, which is understudied according to the authors. This paper is highly descriptive which is not surprising due to the nature of the data, purely based on describing the community structure in these infants in both gut and oral microbiomes. The majority of the outcomes go in line with other preterm infant studies, concluding that the gut microbiome of late and moderate preterm infants are more similar to term infant, highlighting the role of gestational maturity in shaping microbial communities.

We thank the reviewer for this positive feedback.

Minor comments:

1. Figure 4a, probably better to label the late, moderate and extremely preterm infants in the figure for clarity as well as the n - number of samples.

We have added this information to the figure as suggested.

2. For clarity to general readers, perhaps beneficial to define Entry, Term Equivalent Age and corrected gestational age in line 47-48 (page 3).

We added new text for clarification on page 3 line 47 - page 4 line 3: "Entry (the time of enrolment into the study), Term Equivalent Age (TEA; the gestational age at which an infant is considered full-term, i.e., 40 weeks in this study), and at 3, 6, and 12 months corrected gestational age (CA; the age the infant would be if born at full-term, calculated as chronological age minus the number of days born preterm)".

3. Is Figure 1a referring to bacterial richness index Chao1 or absolute species/genus count? It says number of OTUs to me it means absolute species count. Also is data in Figure 1 based on genus or species?

The "number of OTUs" refers to the absolute count of observed species-level OTUs, not a richness index like Chao1. The data in Figure 1a are based on species-level taxonomic resolution. We added new text to the methods section for clarification, see page 5 line 21: "number of OTUs (absolute count of species-level OTUs, representing richness)."

4. Also in Figure 1, could the authors justify why Wilcoxon test was used rather than Kruskal Wallis?

We used the Wilcoxon rank-sum test (with FDR adjustment) because our comparisons were pairwise in nature - specifically, between saliva and stool samples at each time point, and between adjacent time points within each sample type. Since we were not testing for differences across all groups simultaneously, Kruskal-Wallis was not the most appropriate in this context.

5. In lines 10-13 (page 7), it says "all but Bacillota showing significant temporal variation...", this is at phylum level, however there is no single Genus that shows temporal variation between groups? Interpreting microbiome data at genus level will provide more meaningful insights to the readers. Otherwise consider to add a sentence to clarify Bacillota on a genus level.

We apologise for any confusion here. While Bacillota phylum did not show significant overall temporal variation across the first year of life, several genera within Bacillota did vary significantly over time. This was reported in lines 24-28 (page 7): Within each time point, the top five most abundant genera from Bacillota ranged from 3% to 13% in mean relative abundance and included *Streptococcus*, *Veillonella*, *Staphylococcus*, *Enterococcus*, *Clostridium sensu stricto*, *Lactocaseibacillus*, *Blautia*, *Faecalibacterium* and unclassified genera from the families Lachnospiraceae and Ruminococcaceae, many of which showed significant changes between adjacent time points (p values < 0.05; Figure 3b). Overall significant temporal variation at the genus level is additionally recorded in Supplementary Table 2. We have included additional text to clarify this point on lines 41-42 (page 7): "A complete list of genera that showed significant temporal variation is provided in Supplementary Tables 2 and 3".

6. It will be advisable to show relative abundance on genus level in Figure 3 alongside the phylum.

We thank the reviewer for this suggestion. In Figure 3, genus-level relative abundances are already presented via a bubble plot (panel b), which displays the top 25 most abundant genera across all samples, grouped by sample type and time point. Bubble size reflects the average relative abundance within each group, and this is stated in the figure legend. We chose to visualise the genus-level data in this format, rather than using a stacked bar plot as in the phylum-level analysis, in order to highlight significant temporal changes in genus-level relative abundance between adjacent time points, as indicated by the red and green connecting lines.

7. In Figure 2, it may be better to show also the overall NMDS of all the samples, perhaps will visualise better the clustering of the samples according to the microbiome data?

We thank the reviewer for this thoughtful suggestion. We did explore an overall NMDS plot including all samples; however, we found that it was not meaningfully interpretable due to the complexity of the dataset. As in the manuscript, stratifying the NMDS plots by time point allowed for clearer visualisation of sample type-specific clustering and the increasing dissimilarity over time - key aspects of the observed spatiotemporal variability (Figure 2). We therefore chose to present the NMDS plots in this way to most effectively convey these patterns.

Major comments:

1. Authors should attempt to simplify the results and discussion section and focus only on the key findings and data.

We appreciate the reviewer suggesting this but note that Reviewer 2 described the manuscript as “clearly written and well-organized”. We also note that the length of the results (3 pages) and discussion (3 pages; 2 pages of discussion and 1 page for limitations and summary) is standard for these large descriptive studies. Considering this, we have opted to leave the text as presented. Notwithstanding, if suggested by the editors to shorten the manuscript further we would be glad to do so

2. The authors should also make it clear in the beginning that this cohort of infants are probiotic-supplemented (to what I understood) as this impacts the data interpretation and conclusion.

The cohort of late and moderate preterm infants in this study were not probiotic-supplemented. However, extremely preterm infants were supplemented. This is detailed in the methods section on page 4 lines 40-41: “All EP infants received antifungals and probiotics (liquid preparation of *Lactobacillus acidophilus*, *Bifidobacterium bifidum*, and *B infantis* (Labinic, Biofloratech, UK) from date of first full feed)” and in the discussion on page 11 lines 20-21: “This likely reflects the universal use of *Bifidobacterium*-containing probiotics to EP infants in the NICU, while there was no probiotic use for MP/LP infants”.

We have added new text in the results section for additional clarification, see page 8 lines 37-38: “All EP infants, whereas MP and LP infants were not”.

Reviewer #2 (Comments for the Author):

This manuscript by Ahearn-Ford et al. presents a longitudinal analysis of stool and saliva microbiomes from 160 late and moderate preterm (LMPT) infants over the first year of life. The study identifies distinct and maturing microbial communities with increasing diversity over time. Given the relative paucity of data on LMPT infant microbiomes, this work provides a valuable contribution to the field. The manuscript is clearly written and well-organized, with extensive supplementary data. However, several methodological issues require clarification, and some limitations should be more addressed.

We thank the reviewer for noting this valuable contribution to the field and we have addressed the minor points raised.

1. Clarification of Sample Size and Longitudinal Sampling: Although the authors highlight the cohort size (n=160) as a strength, it is important to clarify that no more than 115 samples were collected at any individual time point. A table summarizing the number and percentage of samples available at each time point should be included to improve transparency. Additionally, please specify how many infants contributed longitudinal samples (i.e., samples from more than one time point), as this is critical for interpreting the longitudinal analyses.

We thank the reviewer for this important clarification. As noted in the manuscript (page 6, lines 31-33), we reported the number of infants contributing longitudinal samples for both stool and saliva: “145 (91%) contributed stool samples (median (IQR) of 3 (2-3) samples) and 158 (99%) provided saliva samples (median (IQR) of 3 (1-3) samples)”. We have also reported the decreasing availability of samples at later time points (lines 33-34, page 6: “Less than 50% of enrolled infants contributed samples at 6 months CA and less than 20% at 12 months CA”). To further improve transparency, we have added the number of samples collected at each time point to Supplementary Table 1, as requested. We have opted to report raw numbers rather than percentages in the table to avoid potential confusion (e.g., whether the percentage is calculated based on the total number of samples overall [n = 773] or the total number of enrolled infants [n = 160]).

2. Choice of 16S rRNA Region: The rationale for selecting the V4 region for 16S rRNA sequencing over other commonly used regions (e.g., V1-V2 or V3-V4) is not provided. Please justify this choice, particularly in the context of the oral microbiome, where other variable regions may offer better resolution for specific taxa.

The V4 region was selected to maintain consistency with most studies on early life microbiome development, including previous studies conducted by our group, albeit such work has been primarily focused on stool samples. Thus, given that V4 is widely used and well-supported for gut microbiome profiling, it allowed comparability across studies, including the addition of the publicly available EP cohort to our analysis. While we acknowledge that other regions (e.g., V1-V2 or V3-V4) may offer improved resolution for some oral taxa, using the same region for both stool and saliva samples ensured consistency in data processing and facilitated cross-sample comparisons within this study. We have now added a limitation on page 11 lines 45-46 to highlight this “The V4 region was selected for consistency with most previous work, including the EP cohort, but other variable regions may offer improved resolution for some oral taxa”.

3. Longitudinal Analysis and Statistical Modeling: The division of the dataset into 10 groups to address repeated measures assumes independence of observations, which may not be appropriate for longitudinal data. This may result in inflated type I error rates and biased conclusions. The use of mixed-effects models would be more statistically rigorous and better account for within-subject correlations over time.

We thank the reviewer for their insightful comment. We have revised the Methods section (page 5, lines 10-13) to remove the phrase “and address potential repeated measures,” to avoid implying that independence of observations was assumed throughout the analysis. The revised sentence now reads: “Excluding the analysis with the EP infant group, comparisons considered ten groups that were established according to visit (Entry, TEA, 3, 6 or 12 months CA) and sample type (stool or saliva), to offer a comprehensive examination of spatiotemporal variations in microbial composition”.

We acknowledge that some participants contributed samples at multiple time points, so not all observations are independent, and we appreciate the reviewer’s suggestion regarding the use of mixed-effects models. We initially analysed data between smaller, adjacent time points to capture finer-scale temporal dynamics that might be obscured when examining longer time intervals, focusing on pairwise group-level comparisons of diversity rather than modelling overall diversity trends over time. The incomplete and irregular longitudinal sampling, along with relatively small group sizes (particularly at later time points), limited the feasibility and stability of fitting mixed-effects models for these pairwise diversity comparisons. Instead, we used nonparametric tests with false discovery rate (FDR) correction. Alongside nonparametric pairwise comparisons, we applied linear mixed-effects models for taxonomic analyses to account for repeated measures and subject-level variation, capturing both localised and overall taxonomic changes. Unlike the taxonomic analyses performed with MaAsLin2 - which modelled multiple fixed effects across the *entire dataset* and included subject ID as a random effect - *pairwise* diversity comparisons involved smaller, unbalanced subsets of samples with limited subject overlap between groups.

We recognize that using nonparametric tests under partial dependence may introduce some correlation among observations and potentially inflate Type I error rates. We have revised the manuscript to improve transparency about this limitation on page 11, lines 30-36: “The study suffered attrition over the course of the sampling period due to the COVID-19 pandemic, meaning that longitudinal analysis was conducted primarily at the cohort/population level rather than individual. This may have concealed specific temporal trends at the individual

level. While mixed-effects models were used to account for repeated measures in overall temporal taxonomic analyses, pairwise group-level comparisons between adjacent time points were conducted without modelling repeated measures. This approach may have introduced partial dependence between observations and potentially inflated Type I error rates.”

4. Beta Diversity and Pseudoreplication: Beta diversity was assessed using UniFrac distances and PERMANOVA (via the adonis2 function). However, it is unclear whether permutations were stratified by patient ID. This is essential to avoid pseudoreplication in repeated measures designs. Please clarify this aspect of the analysis. Furthermore, indicate whether NMDS ordinations were performed after adjusting for within-subject correlation.

We thank the reviewer for this important observation. To avoid pseudoreplication, permutation tests were performed within each time point, ensuring that each patient contributed only one sample per permutation. In this way, no patient has duplicate samples in a single test, effectively avoiding the issue of pseudoreplication. NMDS ordinations were conducted on the same dataset without additional adjustment for within-subject correlation, as each analysis at a given time point includes only independent samples.

5. Oral-Gut Microbiome Correlation: Kendall's tau was used to assess correlation between oral and gut microbiomes at each time point. While this is informative, the authors may wish to consider additional approaches such as source-tracking or source-sink analysis to better characterize potential microbial transmission between body sites.

We agree that source-tracking can provide useful insights into the possible origins and contributions of microbial communities across body sites. However, we note several important limitations relevant to our study design and data. First, source-tracking approaches require clear definition of source and sink communities, which in the context of oral and gut microbiomes may be bidirectional rather than unidirectional, complicating interpretation. Second, our dataset consists of genus-level 16S rRNA gene amplicon data, which lacks the strain-level resolution necessary to confidently infer microbial transmission, as many genera encompass diverse species and strains. Third, limited and irregular repeated sampling across participants restricts assessment of transmission dynamics over time. Given these constraints, we focused on correlation analyses and taxonomic overlap as initial exploratory assessments of relationships between oral and gut microbiomes. We believe that more definitive characterisation of microbial transmission would require strain-level metagenomic sequencing and more comprehensive longitudinal sampling. Such analyses are planned for future work on selected subsets of samples.

6. Inclusion of Extremely Preterm (EP) Cohort: The comparison with previously collected microbiome data from seven extremely preterm (EP) infants is limited by small sample size and significant differences in sampling time points, feeding practices, and antibiotic exposure. These limitations substantially reduce the interpretability of the comparison. It may be more appropriate to present this analysis in the supplementary material, along with a clear discussion of the caveats. This would also allow the main figures to focus on the LMPT cohort, which is the core strength of the study. Additionally, please note that the description of Figure 4c is missing from the main text.

We agree that the sample size for EP is unavoidably small and we highlight this on page 11 lines 10-11: “Still, the EP infant group may have been underpowered to detect differences”

Regarding feeding practices and antibiotic exposure as key drivers distinguishing EP from LMPT infants, this simply reflects the real-world differences between these cohorts, which we have acknowledged on page 11, lines 13-16: “Notwithstanding, EP infants are hospitalised and exposed to more microbiome modulating therapies (e.g., antibiotics and probiotics), so distinction of EP from LMPT infants was not unexpected. This suggests hospitalisation over prematurity as the driver of the infant microbiome, but would require a hospitalised LMPT group to confirm and on page 12 lines 8-10: “The study additionally highlighted the significant impact of GA, namely hospitalisation and NICU clinical practice, on early microbial communities”.

While we acknowledge that the small sample size and inherent clinical differences limit the strength of interpretation, this cohort provides one of the first comparison of this nature - particularly for the oral microbiome, which remains underexplored across preterm gestational age groups. Given these considerations, we have chosen to retain the EP infant comparisons within the main manuscript. Notwithstanding, we have further emphasised the limitations in the text to avoid overinterpretation, see page 12, lines 1-3: “Lastly, the limited size and unique clinical course of the EP cohort may confound this analysis, but nonetheless reflects the real-world scenario for EP compared to the LMPT infants.”

Regarding Figure 4c, thank you for pointing out the omission. We have added the following description in the main text to clarify this finding on page 9, lines 29-31: “MP infants had the greatest number of unique stool genera across both time windows, which may partly reflect the increased DOL within this group (Figure 4c)”.

7. Minor Comments: In the Introduction (lines 17-18), please cite relevant literature on the gut microbiome in extremely preterm infants. In the Methods section, clinical data is provided for only 4 of the 7 EP infants. Please include this information for all subjects. For clarity, indicate the number of samples represented in each panel or figure legend of Supplementary Figure 4.

We have now cited additional relevant literature as requested, see page 3, lines 16-18: “To date, most studies have focused on gut microbiome development in very and extremely preterm infants with severe health challenges (Beck et al., 2022; Stewart et al., 2016, 2017; Thänert et al., 2024; Yang et al., 2025; Young et al., 2020).” We have also updated the citations list.

Regarding clinical data for EP infants, we have updated the text on page 4, lines 42-45: “Of the seven EP infants, two had sepsis (culture positive), one had necrotising enterocolitis (medical), and one had both sepsis (culture positive) and necrotising enterocolitis (surgical). The remaining three EP infants had no diagnosed disease, including sepsis or NEC”.

We have also updated the figure legend for Supplementary Figure 4, which now reads: “For each time point, both stool and saliva samples were available for the same number of patients: 77 pairs at Entry, 98 pairs at TEA, 74 pairs at 3mo, 61 pairs at 6mo, and 27 pairs at 12mo (i.e., 2 samples per patient per time point)”.

Re: mSystems00667-25R1 (Spatiotemporal development of late and moderate preterm infant gut and oral microbiomes and impact of gestational age on early colonisation)

Dear Prof. Christopher J Stewart:

Your manuscript has been accepted, and I am forwarding it to the ASM production staff for publication. Your paper will first be checked to make sure all elements meet the technical requirements. ASM staff will contact you if anything needs to be revised before copyediting and production can begin. Otherwise, you will be notified when your proofs are ready to be viewed.

Sincerely,
David Cleary
Editor
mSystems